# Channel-Spatial Support-Query Cross-Attention for Fine-Grained Few-Shot Image Classification

## ABSTRACT

Few-shot fine-grained image classification aims to use only few labelled samples to successfully recognize subtle sub-classes within the same parent class. This task is extremely challenging, due to the co-occurrence of large inter-class similarity, low intra-class similarity, and only few labelled samples. In this paper, to address these challenges, we propose a new Channel-Spatial Cross-Attention Module (CSCAM), which can effectively drive a model to extract discriminative fine-grained feature representations with only few shots. CSCAM collaboratively integrates a channel cross-attention module and a spatial cross-attention module, for the attentions across support and query samples. In addition, to fit for the characteristics of fine-grained images, a support averaging method is proposed in CSCAM to reduce the intra-class distance and increase the inter-class distance. Extensive experiments on four few-shot fine-grained classification datasets validate the effectiveness of CSCAM. Furthermore, CSCAM is a plug-and-play module, conveniently enabling effective improvement of state-of-the-art methods for few-shot fine-grained image classification.

## KEYWORDS

Few-shot learning, Fine-grained image classification, Channel cross-attention, Spatial cross-attention

## 1 INTRODUCTION

Few-shot fine-grained image classification aims to recognize subtle sub-classes within the same parent class (e.g., bird species [35], car models [12]), with only few labeled samples for training. In addition to the intrinsic challenge of few labeled samples, there are two more major challenges co-occurring in few-shot fine-grained image classification: large inter-class similarity, and low intra-class similarity [19, 36].

Therefore, technically speaking, few-shot fine-grained image classification needs to tackle challenges from two frontiers, fine grains and few shots. On the one hand, fine-grained image classification requires attention to small and hard-to-explore discriminative feature regions, which is challenging even for advanced few-shot image classification methods such as Proto [28] and FRN [38], as shown in Figure 1. On the other hand, most of existing fine-grained image classifiers rely on a large number of labeled samples for training, which is unavailable under the few-shot setting. Hence,

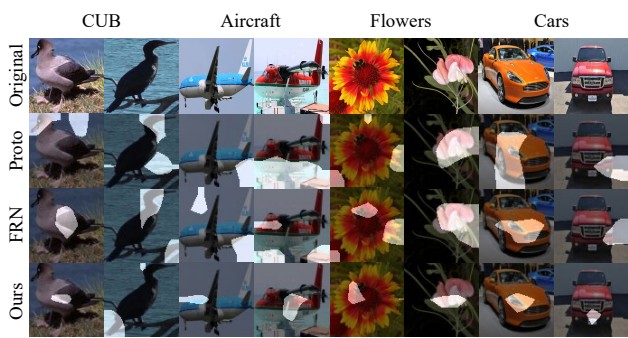

**Figure 1: Queries from four fine-grained datasets (CUB, Aircraft, Flowers, Cars). The Grad-CAM visualization showcases areas of interest localised by Proto [28], FRN [38], and our method. Ours offers more focused discriminative areas.**

how to deal with few-shot fine-grained image classification task is an extremely challenging topic.

Metric learning, which classifies query images by comparing the distance between query features and support features, is widely used in few-shot learning. For example, Proto [28] uses the cosine similarity; Relation [30] learns a metric; FRN [38], TDM [14], BiFRN [40] and BSFA [44] use reconstructed features. Many metric learning networks can be diagrammed as Figure 2(a).

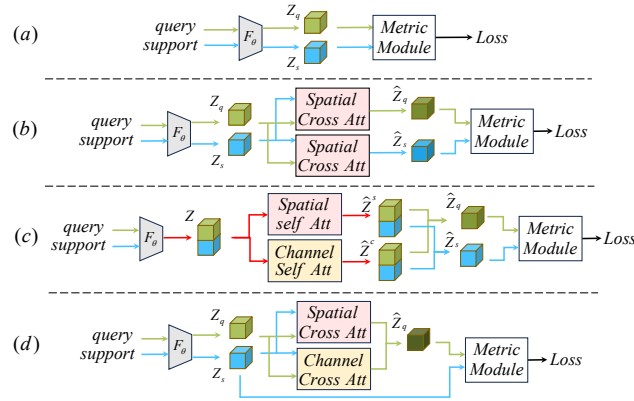

**Figure 2: Different attention mechanisms for metric-based few-shot learning. (a) Without attention, e.g., Proto [28] and FRN [38]. (b) With spatial cross-attention, e.g., CAM [9] and CAD [4]. (C) With channel and spatial self-attentions, e.g., MattML [47]. (d) Our method with collaborative channel-spatial support-query cross-attentions, which can generate more focused and discriminative features to improve the performance of few-shot fine-grained image classification.**

Attention mechanisms have also been explore for few-shot learning. For example, Hou et al. [9] and Chikontwe et al. [4] use spatial cross-attention to implicitly reweight the underlying spatial map to focus on relevant target regions, which can be combined with metric learning and diagrammed as in Figure 2(b). Recently, people use the self-attention mechanism in both channel and space to learn better corresponding relation, which can be diagrammed as Figure 2(c), for example, Zhu et al. [47] and Xu et al. [42]. However, none of them explore the cross-attention in both channel and space.

Therefore, this paper considers the cross-attentions between support and query features from both spatial and channel perspectives, and proposes the *channel-spatial cross-attention module (CSCAM)* as diagrammed in Figure 2(d), which can improve the state-of-the-art performance in few-shot fine-grained image classification.

The proposed CSCAM contains two sub-modules: a channel cross-attention module (CCAM), and a spatial cross-attention module (SCAM). CCAM explores the dependence between the query and support samples along the feature channel dimension; SCAM explores the spatial dependence between the query and support samples. Finally, the outputs of these two sub-modules are mixed to simultaneously enhance the feature representations in the channel and spatial dimensions and generate more discriminative features.

In addition, to fit for the characteristics of fine-grained images, we propose a support averaging method to further adjust the weight of the cross-attention score to expand the inter-class distance and reduce the intra-class distance.

In summary, the novelties of our CSCAM are twofold:

First, unlike existing methods, we consider the cross-attentions from both channel and space perspectives, fully leveraging the spatial and channel cross-information between support and query.

Secondly, different from the existing attention methods that usually generate class prototypes before attention or stack multi-dimensional features after attention, CSCAM averages the attention scores of all support features, reducing the between-class similarity and increasing the within-class similarity, as shown by ablation studies and visualisation.

Both novelties makes CSCAM fit for few-shot fine-grained image classification. Extensive experiments including ablation studies show that CSCAM achieves excellent performance on few-shot fine-grained image classification tasks. Moreover, we note that CSCAM is a plug-and-play module that can be integrated without restricting the choice of embedding module and metric module.

## 2 RELATED WORK

### 2.1 Metric-based Few-shot Fine-grained Classification

In few-shot learning, metric-based methods have been widely used because of their simplicity and efficiency [18]. Well-established methods in this category include Proto [28], which evaluates the Euclidean distance between a query sample and a class prototype, and Relation [30], which learns a metric. In recent years, feature reconstruction-based approaches have also demonstrated promising results. TDM [14] highlights the information of different channels. BiFRN [40] proposes a self-reconstruction module and a bidirectional reconstruction module to enlarge the inter-class distance and reduce the intra-class distance. BSFA [44] proposes a

two-stage framework that incorporates background suppression and foreground alignment to localize foreground objects and mitigate background interference. However, the problems with these methods include insufficient mining of channel information and insufficient capture of the cross-information between support and query samples, hence the features generated by them are still not strong enough for few-shot fine-grained image classification.

Different from the aforementioned methods, our proposed CSCAM mixes channel and spatial cross-attentions to fully leverage the spatial and channel cross-information.

### 2.2 Attention Mechanisms for Few-shot Fine-grained Classification

The objective of attention mechanisms is to highlight important local regions, thereby enhancing the discriminative nature of the reweighted features. In recent years, various attention mechanisms have been explored in few-shot learning. For example, Hou et al. [9] employed spatial cross-attention between support features and query features to reweight key regions of the target. Similarly, Chikontwe et al. [4] utilized spatial cross-attention from the mixed-set to the prototype, placing greater emphasis on critical regions and generating adaptive reweighted features. However, these two methods only use cross-spatial attentions and are not for fine-grained classification. For few-shot fine-grained image classification, Zhu et al. [47] utilized two CBAM modules to adaptively attend to the discriminative parts from both channel and spatial perspectives. Xu et al. [42] propose two dual branches incorporating both hard and soft attention. However, these two attention methods do not consider the channel and spatial cross-attentions between support and query features simultaneously.

Different from the above methods, we construct and fuse cross-attentions from both channel and spatial perspectives. In addition, to fit the characteristics of fine-grained images, we propose a new support averaging method for channel and spatial cross-attentions, which reduces the weight of similar regions from different classes and increases the weight of similar regions within the same class.

## 3 METHOD

### 3.1 Problem Setting

In standard few-shot classification, both the training and test stages consist of multiple episodes. Each episode is composed of randomly sampled $C$ classes, and within each class, there are $K$ support images and $U$ query images. This is referred to as a $C$-way $K$-shot episode. In each episode, the model is provided with the $K$ labeled images from each of the $C$ classes and is tasked with correctly classifying the $U$ unlabeled images. The model's performance is evaluated based on its ability to accurately classify the unlabeled images in these episodes. Given a dataset $\mathcal{D} = \{(x_i, y_i), y_i \in \mathcal{Y}\}$, where $\mathcal{Y}$ is the label set. It is divided into three parts, that is, $\mathcal{D}_{train} = \{(x_i, y_i), y_i \in \mathcal{Y}_{train}\}$, $\mathcal{D}_{val} = \{(\bar{x}_i, \bar{y}_i), \bar{y}_i \in \mathcal{Y}_{val}\}$ and $\mathcal{D}_{test} = \{(\widetilde{x}_i, \widetilde{y}_i), \widetilde{y}_i \in \mathcal{Y}_{test}\}$, where $x_i, \bar{x}_i, \widetilde{x}_i$ and $y_i, \bar{y}_i, \widetilde{y}_i$ are the original image and class label of the $i^{th}$ image on $\mathcal{D}_{train}, \mathcal{D}_{val}, \mathcal{D}_{test}$, respectively. The label space of the training labels $\mathcal{Y}_{train}$, validation labels $\mathcal{Y}_{val}$ and test labels $\mathcal{Y}_{test}$ are non-overlapping, i.e., $\{\mathcal{Y}_{train} \cap \mathcal{Y}_{val}\} = \phi$, $\{\mathcal{Y}_{train} \cap \mathcal{Y}_{test}\} = \phi$, and $\{\mathcal{Y}_{val} \cap \mathcal{Y}_{test}\} = \phi$.

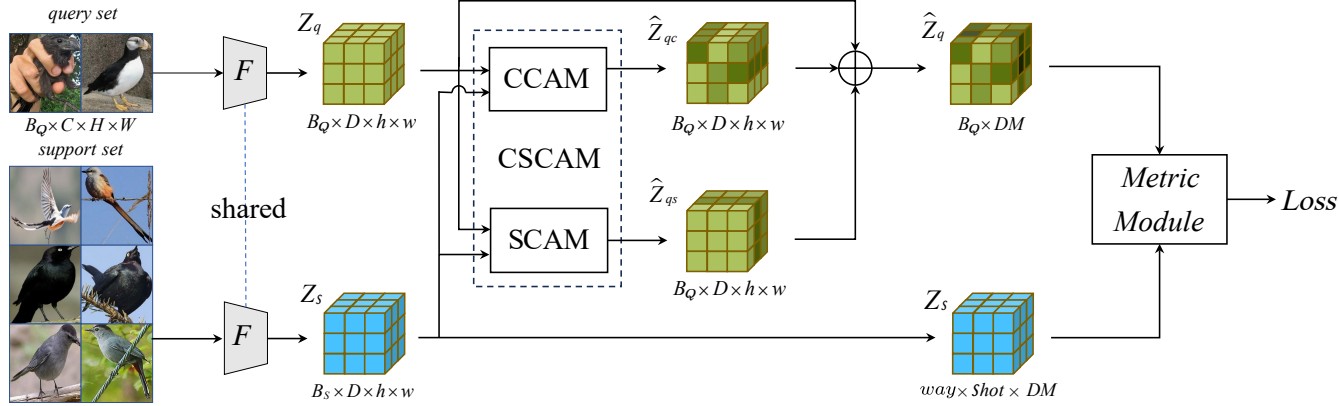

**Figure 3: Diagram of our proposed method. We first use a shared backbone $F_\theta$ to extract globally pooled features, then use a channel-spatial cross-attention module (CSCAM) to refine the features of the query set samples. The query embedding $Z_q$ and the support embedding $Z_s$ are fed into CSCAM, which re-weights $Z_q$ to produce $\widehat{Z}_{qc}$ and $\widehat{Z}_{qs}$ in channel and spatial aspects, respectively. After that, we collaboratively mix the channel and spatial re-weighted query features to obtain $\widehat{Z}_q$. Finally, we feed $\widehat{Z}_q$ and $Z_s$ into the metric module for image classification. Notation: *way* represents the number of classes; *shot* is the number of images in each class; $B_S$ and $B_Q$ are the sizes of support sets and query sets, respectively; $C$ represents the number of channels of the images; $D$ is the number of channels of the features; $H, W$ represent the height and width of the images; $h, w$ represent the height and width of the features; and $M$ is equal to $h \times w$.**

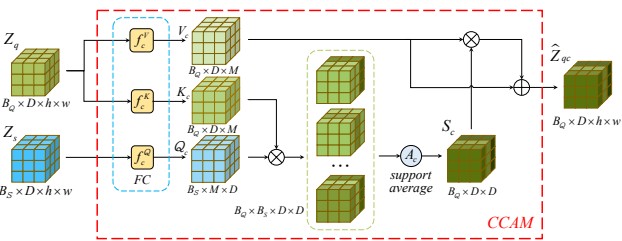

**Figure 4: Diagram of the Channel-Cross Attention Module (CCAM). The initial support embeddings $Z_s$ and query embeddings $Z_q$ are fed into CCAM; $Z_q$ is projected as $K_c$ and $V_c$, and $Z_s$ as $Q_c$. Duplicate $B_Q$ copies of $Q_c$ and $B_S$ copies of $K_c$ and compute channel attention scores. Then, the *support average* $S_c$ of the attention scores re-weights query features $V_c$ to obtain $\widehat{Z}_{qc}$. Notation: $B_S$ and $B_Q$ denote the cardinalities of support sets and query sets, respectively; $D$ is the number of channels; $h, w$ represent the height and width of the feature; $M$ is equal to $h \times w$; and $f_c^Q, f_c^K, f_c^V$ are fully-connected (FC) layers for feature projections. Note: The spatial cross-attention module (SCAM), with its diagram omitted here, is structurally similar to CCAM, but with feature projections in the channel dimension replaced by $f_s^Q, f_s^K, f_s^V$ in the spatial dimension, hence the resulting $Q_s, K_s, V_s, S_s,$ and $\widehat{Z}_{qs}$.**

In few-shot classification, the objective is to improve the performance of $C$-way $K$-shot classification on the test dataset $\mathcal{D}_{test}$ by leveraging knowledge from the training dataset $\mathcal{D}_{train}$ and selecting optimal model weights through the validation dataset $\mathcal{D}_{val}$.

During episodic training, a task $\mathcal{T}$ is formed by randomly sampling $C$ classes from $\mathcal{D}_{train}$, with each class having $K$ randomly selected labeled (*support*) samples $\mathcal{S} = \{(x_s, y_s)\}_{s=1}^{C \times K}$ and $U$ randomly sampled unlabeled (*query*) samples $\mathcal{Q} = \{(x_q, y_q)\}_{q=1}^{C \times U}$. Similarly, tasks $\bar{\mathcal{T}}$ and $\widetilde{\mathcal{T}}$ are defined on $\mathcal{D}_{val}$ and $\mathcal{D}_{test}$ for validation and test scenarios, respectively. The training process on $\mathcal{D}_{train}$ is analogous to the prediction process on $\mathcal{D}_{test}$.

## 3.2 Channel-Spatial Cross-Attention Module (CSCAM)

*3.2.1 Overview.* The diagram of our proposed method including the Channel-Spatial Cross-Attention Module (CSCAM) is shown in Figure 3. CSCAM consists of two attention modules named Channel Cross-Attention Module (CCAM) and Spatial Cross-Attention Module (SCAM). Different from existing attention methods, CSCAM collaborates cross-attentions from both channel and spatial perspectives. To fit for the characteristics of fine-grained images, a support averaging method is proposed on the basis of the two-aspect attentions, with the attention scores further adjusted to enlarge the inter-class distance and reduce the intra-class distance. Specifically, for each query feature, all support features are averaged to capture the relationship between each query feature and all support features, reducing the weight of similar regions from different classes and increasing the weight of similar regions within the same class.

To obtain more discriminative features, we introduce a cross-attention mechanism from both spatial and channel perspectives, to re-weight the query features by considering the relevant features between the support features and the query features. Let $D$ be the number of channels of the feature, $M$ be the resolution ($h \times$

$w$) of the feature, and $B_s$ and $B_Q$ be the amount of data of the support features and query features, respectively. Formally, let $Z_s \in \mathbb{R}^{B_s \times D \times M}$ and $Z_q \in \mathbb{R}^{B_Q \times D \times M}$ be the globally pooled features extracted by backbone $F_\theta$. Then, query embeddings $Z_q$ and support embeddings $Z_s$ are further fed to our channel-spatial cross-attention module to generate re-weighted query features. Finally, we feed the re-weighted query features $\widehat{Z}_q$ and support features $Z_s$ into the metric module.

*3.2.2 Channel Cross-Attention Module (CCAM).* Following the work of Vaswani et al. [34], we employ an attention function to re-weight $V$ using the similarity between $Q$ and $K$:

$$\varphi(Q, K, V) = \text{softmax}(\frac{QK^T}{\sqrt{D}}V). \tag{1}$$

In our CCAM, as shown in Figure 4, through feature projections using fully-connected (FC) layers $f_c^Q, f_c^K, f_c^V$ from the channel perspective, the projections of the query embedding $Z_q$ are denoted by $K_c$ and $V_c$ and the projection of the support embedding $Z_s$ by $Q_c$:

$$Q_c = f_c^Q(Z_s), K_c = f_c^K(Z_q), V_c = f_c^V(Z_q). \tag{2}$$

Then, we duplicate $B_Q$ copies of $Q_c$ and $B_S$ copies of $K_c$, to facilitate the subsequent operation, where the support average attention score $S_c$ is obtained by performing the support averaging operation $A_c$ on the attention scores as

$$S_c = A_c(Q_c, K_c) = \frac{1}{B_S} \sum_{i=1}^{B_S} \text{softmax}(\frac{Q_c \times (K_c)^T}{\sqrt{D'}}), \tag{3}$$

where $D'$ is the number of channels after projection.

The obtained support average attention score $S_c$ is then used to weight $V_c$:

$$\varphi(Q_c, K_c, V_c) = S_c \times V_c = A_c(Q_c, K_c) \times V_c. \tag{4}$$

*3.2.3 Spatial Cross-Attention Module (SCAM).* The architecture of SCAM is similar to that of CCAM in Figure 4, hence the similarity between the computation of SCAM and that of CCAM in the section above.

In the end, similar to Equation (4), the obtained support average attention score $S_s$ is used to weight the $V_s$:

$$Q_s = f_s^Q(Z_s), K_s = f_s^K(Z_q), V_s = f_s^V(Z_q), \tag{5}$$

$$S_s = A_s(Q_s, K_s) = \frac{1}{B_S} \sum_{i=1}^{B_S} \text{softmax}(\frac{Q_s \times (K_s)^T}{\sqrt{D'}}), \tag{6}$$

$$\varphi(Q_s, K_s, V_s) = S_s \times V_s = A_s(Q_s, K_s) \times V_s. \tag{7}$$

*3.2.4 Integration of CCAM and SCAM.* The re-weighted query features obtained in channel $\hat{Z}_{qc}$ and spatially $\hat{Z}_{qs}$ are mixed to obtain a more discriminative query feature $\hat{Z}_q$. In addition, to prevent the attention from focusing on non-critical areas, which causes the weight of key locations to become extremely small and causes overfitting [7], we retain $V_c$ and $V_s$. This is represented by summing the original query features:

$$\hat{Z}_{qc} = \varphi(Q_c, K_c, V_c) + V_c,$$
$$\hat{Z}_{qs} = \varphi(Q_s, K_s, V_s) + V_s, \tag{8}$$

$$\hat{Z}_q = f^O(m_1 \times \hat{Z}_{qc} + m_2 \times \hat{Z}_{qs}) + Z_q, \tag{9}$$

where $m_1, m_2$ with the initial value of 0.5 can be learnable parameters, representing the weights of the channel and space, and $f^O$ is a fully-connected (FC) layer.

The innovation of CSCAM is to capture the channel and spatial cross-attention between query features and support features simultaneously. Also, to fit for the characteristics of fine-grained images, the support averaging method is proposed to adjust the attention scores from both channel and spatial perspectives. Specifically, the whole support set of the attention scores is averaged to increase the weight of regions with high similarity within the same class, and reduce the weight of regions with high similarity from different classes, hence enlarging the inter-class distance and reducing the intra-class distance to generate discriminative features more suitable for few-shot fine-grained image classification tasks.

## 3.3 Plug-and-Play with Existing Methods

Our proposed module CSCAM can be readily integrated into existing metric-based few-shot learning models, for example, Proto [28], Relation [30], FRN [38] and TDM [14] (inductive only), using their metric module and loss functions. Here, for illustrative purposes, we use FRN [38] as an example to outline the entire network training process.

The output of the convolutional feature extractor for $x_s, x_q$ are feature map $Z_s \in \mathbb{R}^{M \times D}, Z_q \in \mathbb{R}^{M \times D}$, where $M$ denotes the spatial resolution (height × width) of the feature map, and $D$ denotes the number of channels. We obtain the weighted query features $\hat{Z}_q$ after feeding $Z_s, Z_q$ into our CSCAM:

$$\hat{Z}_q = CSCAM(Z_s, Z_q). \tag{10}$$

For each class $c \in C$, pool all features from the $K$ support images into a single matrix of support features $Z_{sc}$, and calculate the final probability distribution:

$$P(y_q = c | x_q) = \frac{\exp(-d \langle \hat{Z}_q, Z_{sc} \rangle)}{\sum_{c' \in C} \exp(-d \langle \hat{Z}_q, Z_{sc'} \rangle)}, \tag{11}$$

where $d$ denotes the metric module.

Following FRN [38], it optimizes the network by the cross-entropy loss $loss_{cross}$ and the auxiliary loss $loss_{aux}$ (details in FRN [38]):

$$loss_{cross} = -\frac{1}{B_Q} \sum_{q=0}^{B_Q} y_q^\top \log(P(y_q | x_q)). \tag{12}$$

$$loss = loss_{cross} + loss_{aux}, \tag{13}$$

where $B_Q$ is the number of query images, and $y_q$ is the one-hot vector that predicts the label of the $q$-th query image.

In short, our CSCAM, only requiring the input support features and query features, is a plug-and-play module.

## 4 EXPERIMENTAL ANALYSIS

### 4.1 Datasets

We conduct all the experiments on four benchmark datasets:

*CUB-200-2011* [35] (CUB): 11,788 images from 200 bird species;

*FGVC-Aircraft* [22] (Aircraft): 10,000 images from 100 aircraft species;

*Flowers-102* [24] (Flowers): 102 categories of common flowers;

*Stanford-Cars* [12] (Cars): 16,185 images of 196 classes of cars.

**Table 1: Performance of five-way few-shot classification on the CUB-200-2011 (CUB), FGVC-Aircraft (AIRCRAFT), 102 Flowers (FLOWERS), and Stanford-Cars (CARS) datasets using the ResNet-12 and Conv-4 backbones, respectively.**

| ResNet-12 | CUB | | Aircraft | | Flowers | | Cars | |
|---|---|---|---|---|---|---|---|---|
| | 1-shot | 5-shot | 1-shot | 5-shot | 1-shot | 5-shot | 1-shot | 5-shot |
| Proto (NeurIPS'17) [28]† | 79.64 ± 0.20 | 91.15 ± 0.11 | 86.57 ± 0.18 | 93.51 ± 0.09 | 75.41 ± 0.22 | 89.46 ± 0.14 | 82.29 ± 0.20 | 93.11 ± 0.10 |
| Relation (CVPR'18) [30] † | 63.94 ± 0.92 | 77.87 ± 0.64 | 64.24 ± 1.03 | 77.56 ± 0.66 | 65.93 ± 1.00 | 85.10 ± 0.63 | 65.77 ± 0.99 | 84.29 ± 0.60 |
| Baseline++ (ICLR'19) [3] ◇ | 64.62 ± 0.98 | 81.15 ± 0.61 | 74.51 ± 0.90 | 88.06 ± 0.44 | 69.03 ± 0.92 | 85.72 ± 0.63 | 67.92 ± 0.92 | 84.17 ± 0.58 |
| DeepEMD (CVPR'20) [46]◇ | 71.11 ± 0.31 | 86.30 ± 0.19 | 69.86 ± 0.30 | 85.17 ± 0.28 | 70.00 ± 0.35 | 83.63 ± 0.26 | 73.30 ± 0.29 | 88.37 ± 0.17 |
| VFD (ICCV'21) [41] ◇ | 79.12 ± 0.83 | 91.48 ± 0.39 | 76.88 ± 0.85 | 88.77 ± 0.46 | 76.20 ± 0.92 | 89.90 ± 0.53 | 77.52 ± 0.85 | 90.76 ± 0.46 |
| RENet (ICCV'21) [11] ◇ | 79.49 ± 0.44 | 91.11 ± 0.24 | 82.04 ± 0.41 | 90.50 ± 0.24 | 79.91 ± 0.42 | 92.33 ± 0.22 | 79.66 ± 0.44 | 91.95 ± 0.22 |
| MixFSL (ICCV'21) [1] ◇ | 67.87 ± 0.94 | 82.18 ± 0.66 | 60.55 ± 0.86 | 77.57 ± 0.69 | 72.60 ± 0.91 | 86.52 ± 0.65 | 58.15 ± 0.87 | 80.54 ± 0.63 |
| FRN (CVPR'21) [38] † | 83.11 ± 0.19 | 92.49 ± 0.11 | 87.53 ± 0.18 | 93.98 ± 0.09 | 81.07 ± 0.20 | 92.52 ± 0.11 | 85.91 ± 0.18 | 94.52 ± 0.09 |
| CAD (CVPR'22) [4] ✱ | 82.95 ± 0.67 | 90.80 ± 0.51 | - | - | - | - | - | - |
| AGPF (PR'22) [31] ✱ | 78.73 ± 0.84 | 89.77 ± 0.47 | - | - | - | - | 85.34 ± 0.74 | 94.79 ± 0.35 |
| HelixFormer (MM'22) [45] † | 81.66 ± 0.30 | 91.83 ± 0.17 | 75.79 ± 0.23 | 83.03 ± 0.16 | 63.30 ± 0.26 | 66.96 ± 0.22 | 79.40 ± 0.43 | 92.26 ± 0.15 |
| TDM (CVPR'22) [14] † | 82.41 ± 0.19 | 92.37 ± 0.10 | 87.96 ± 0.17 | 94.20 ± 0.08 | 82.41 ± 0.19 | 93.42 ± 0.10 | 86.77 ± 0.17 | 95.94 ± 0.07 |
| BSFA (IEEE TCSVT'23) [44] † | 82.27 ± 0.46 | 90.76 ± 0.26 | 87.85 ± 0.35 | 94.93 ± 0.14 | 74.48 ± 0.54 | 86.05 ± 0.36 | **88.93 ± 0.38** | 95.20 ± 0.20 |
| IDEAL-clean (TPAMI'23) [2] † | 77.56 ± 0.86 | 88.87 ± 0.51 | 61.37 ± 0.92 | 82.51 ± 0.55 | 74.39 ± 0.93 | 87.29 ± 0.61 | 74.02 ± 0.89 | 89.98 ± 0.50 |
| BiFRN (AAAI'23) [40] † | 83.08 ± 0.19 | 93.33 ± 0.10 | 86.88 ± 0.17 | 93.74 ± 0.09 | 80.62 ± 0.20 | 92.54 ± 0.11 | 87.98 ± 0.16 | **96.66 ± 0.06** |
| C2-Net (AAAI'24) [21] † | 83.37 ± 0.42 | 92.20 ± 0.23 | 87.98 ± 0.39 | 93.96 ± 0.20 | 80.86 ± 0.46 | 91.54 ± 0.27 | 84.42 ± 0.43 | 92.72 ± 0.23 |
| **Proto+CSCAM (Ours)** | 81.69 ± 0.20 | 91.01 ± 0.11 | 87.45 ± 0.17 | 93.68 ± 0.09 | 80.36 ± 0.20 | 91.11 ± 0.12 | 85.73 ± 0.18 | 93.26 ± 0.10 |
| **Relation+CSCAM (Ours)** | 71.14 ± 0.95 | 85.15 ± 0.58 | 67.72 ± 1.05 | 78.24 ± 0.63 | 69.62 ± 0.96 | 85.17 ± 0.66 | 70.08 ± 0.99 | 85.63 ± 0.64 |
| **FRN+CSCAM (Ours)** | **84.00 ± 0.18** | **93.52 ± 0.10** | **88.01 ± 0.17** | **95.06 ± 0.07** | 82.35 ± 0.19 | **93.87 ± 0.10** | 86.24 ± 0.18 | 95.55 ± 0.08 |
| **TDM+CSCAM (Ours)** | 83.34 ± 0.19 | 92.98 ± 0.10 | 87.84 ± 0.17 | 94.37 ± 0.08 | **82.50 ± 0.19** | 93.57 ± 0.10 | 86.86 ± 0.17 | 95.63 ± 0.08 |

| Conv-4 | CUB | | Aircraft | | Flowers | | Cars | |
|---|---|---|---|---|---|---|---|---|
| | 1-shot | 5-shot | 1-shot | 5-shot | 1-shot | 5-shot | 1-shot | 5-shot |
| Proto (NeurIPS'17) [28] † | 61.82 ± 0.23 | 83.37 ± 0.15 | 50.90 ± 0.22 | 80.65 ± 0.15 | 64.23 ± 0.23 | 84.94 ± 0.16 | 48.42 ± 0.22 | 71.38 ± 0.18 |
| Relation (CVPR'18) [30] † | 63.94 ± 0.92 | 77.87 ± 0.64 | 61.73 ± 0.98 | 75.96 ± 0.72 | 69.50 ± 0.96 | 83.91 ± 0.63 | 46.04 ± 0.91 | 68.52 ± 0.78 |
| Baseline++ (ICLR'19) [3] ◇ | 62.36 ± 0.84 | 79.08 ± 0.61 | 58.38 ± 0.83 | 77.62 ± 0.60 | 70.54 ± 0.84 | 86.63 ± 0.58 | 46.82 ± 0.76 | 68.20 ± 0.72 |
| DN4 (CVPR'19) [15] ✱ | 57.45 ± 0.89 | 84.41 ± 0.58 | 68.41 ± 0.91 | 87.48 ± 0.49 | 70.44 ± 0.95 | 89.45 ± 0.52 | 34.12 ± 0.68 | 87.47 ± 0.47 |
| DSN (CVPR'20) [27] ◇ | 71.57 ± 0.92 | 83.51 ± 0.60 | 66.30 ± 0.87 | 79.00 ± 0.61 | 67.71 ± 0.92 | 84.58 ± 0.70 | 48.16 ± 0.86 | 60.77 ± 0.75 |
| BSNet (D&C) (IEEE TIP'20) [17] ✱ | 62.84 ± 0.95 | 85.39 ± 0.56 | 56.51 ± 1.09 | 70.80 ± 0.81 | 66.60 ± 1.04 | 80.42 ± 0.75 | 40.89 ± 0.77 | 86.88 ± 0.50 |
| MattML (IJCAI'20) [47] ✱ | 66.29 ± 0.56 | 80.34 ± 0.30 | - | - | - | - | 66.11 ± 0.54 | 82.80 ± 0.28 |
| MixFSL (ICCV'21) [1] ◇ | 53.61 ± 0.88 | 73.24 ± 0.75 | 44.89 ± 0.75 | 62.81 ± 0.73 | 68.01 ± 0.90 | 85.10 ± 0.62 | 44.56 ± 0.80 | 59.63 ± 0.79 |
| FRN (CVPR'21) [38] † | 73.46 ± 0.21 | 88.13 ± 0.13 | 69.29 ± 0.22 | 83.94 ± 0.13 | 73.60 ± 0.22 | 88.69 ± 0.14 | 64.03 ± 0.22 | 84.02 ± 0.13 |
| TDM (CVPR'22) [14] † | 74.39 ± 0.21 | 88.89 ± 0.13 | 69.90 ± 0.23 | 83.34 ± 0.15 | 70.66 ± 0.24 | 85.14 ± 0.17 | 65.89 ± 0.22 | 82.45 ± 0.15 |
| DUAL ATT-NET (AAAI'22) [42] ✱ | 72.89 ± 0.50 | 86.60 ± 0.31 | - | - | - | - | 70.21 ± 0.50 | 85.55 ± 0.31 |
| CAML (WACV'23) [29] ✱ | 59.71 ± 1.46 | 73.09 ± 0.73 | 57.55 ± 1.37 | 72.88 ± 0.64 | - | - | - | - |
| BSFA (IEEE TCSVT'23) [44] † | 68.16 ± 0.52 | 82.41 ± 0.35 | 61.17 ± 0.49 | 76.96 ± 0.36 | 66.84 ± 0.59 | 79.34 ± 0.47 | 49.98± 0.48 | 67.52 ± 0.44 |
| IDEAL-clean (TPAMI'23) [2] † | 69.93 ± 0.89 | 81.67 ± 0.69 | 52.56 ± 0.83 | 80.36 ± 0.69 | 72.25 ± 0.90 | 86.43 ± 0.70 | 52.64 ± 0.91 | 70.28 ± 0.69 |
| BiFRN (AAAI'23) [40] † | 76.52 ± 0.21 | 89.75 ± 0.11 | 75.72 ± 0.21 | 86.91 ± 0.12 | 71.49 ± 0.23 | 85.02 ± 0.16 | 72.24 ± 0.21 | 87.39 ± 0.12 |
| **Proto+CSCAM (Ours)** | 72.33 ± 0.21 | 87.18 ± 0.13 | 69.96 ± 0.23 | 85.24 ± 0.13 | 71.20 ± 0.23 | 85.43 ± 0.16 | 66.58 ± 0.23 | 83.25 ± 0.15 |
| **Relation+CSCAM (Ours)** | 65.50 ± 1.02 | 79.10 ± 0.66 | 53.04 ± 0.95 | 67.22 ± 0.79 | 66.02 ± 1.04 | 80.72 ± 0.73 | 45.63 ± 0.87 | 71.50 ± 0.79 |
| **FRN+CSCAM (Ours)** | 77.68 ± 0.20 | **89.88 ± 0.12** | 76.12 ± 0.20 | 88.02 ± 0.12 | **74.29 ± 0.22** | **88.70 ± 0.14** | 71.44 ± 0.21 | 86.44 ± 0.13 |
| **TDM+CSCAM (Ours)** | **77.81 ± 0.21** | 89.60 ± 0.12 | **78.84 ± 0.20** | **88.89 ± 0.11** | 74.08 ± 0.22 | 88.63 ± 0.14 | **73.27 ± 0.21** | **87.81 ± 0.12** |

✱: results from BSNet [17].    ◇: results from LCCRN [16].    ✱: results reported in the official papers.
†: reproduced results from using the officially published code.

For each dataset, following the setting in [38], we divide the data into training set $\mathcal{D}_{train}$, validation set $\mathcal{D}_{val}$ and test set $\mathcal{D}_{test}$ in a ratio of 2:1:1. All images are resized to 84×84. Dataset preprocessing also follows [38]; each image in CUB is cropped to a manually annotated bounding box, while original images is used from the other three datasets.

## 4.2 Implementation Details

In our experiments, we set the training epoch to 1,200, the initial learning rate to 0.1, and the weight decay to 5e-4. The learning rate is decreased after every 400 epochs, and we verify the model's performance every 20 epochs to monitor its progress during training.

During the test stage, we use the trained model to classify 10,000 test images. The average classification accuracy of the model on these test images is taken as the performance metric.

**Table 2: Five-way few-shot classification performance on the *CUB-200-2011* (CUB) and *Stanford-Cars* (Cars) datasets for the Conv-4 and the ResNet-12 backbone.**

| Attention module | CUB (Conv-4) | | Cars (Conv-4) | | CUB (ResNet-12) | | Cars (ResNet-12) | |
|---|---|---|---|---|---|---|---|---|
| | *1-shot* | *5-shot* | *1-shot* | *5-shot* | *1-shot* | *5-shot* | *1-shot* | *5-shot* |
| Proto [28] | 63.21 ± 0.23 | 83.88 ± 0.15 | 48.70 ± 0.22 | 72.18 ± 0.18 | 78.56 ± 0.20 | 90.39 ± 0.11 | 82.29 ± 0.20 | 93.11 ± 0.10 |
| +SE [10] | 68.27 ± 0.22 | 86.20 ± 0.14 | 64.15 ± 0.23 | 81.70 ± 0.15 | 77.14 ± 0.21 | 90.67 ± 0.11 | 80.83 ± 0.20 | 93.82 ± 0.09 |
| +CBAM [39] | 69.89 ± 0.22 | 86.29 ± 0.14 | 64.43 ± 0.24 | 82.34 ± 0.15 | 78.12 ± 0.20 | 90.95 ± 0.11 | 80.00 ± 0.21 | 93.39 ± 0.09 |
| +DA[5] | 68.18 ± 0.22 | 86.28 ± 0.14 | 59.63 ± 0.22 | 81.08 ± 0.16 | 75.95 ± 0.21 | 90.85 ± 0.11 | 80.23 ± 0.21 | 93.88 ± 0.09 |
| +CAM [9] | 68.43 ± 0.22 | 86.41 ± 0.13 | 61.88 ± 0.22 | 81.60 ± 0.15 | 76.53 ± 0.21 | 90.50 ± 0.11 | 82.43 ± 0.20 | 93.83 ± 0.09 |
| +Triplet [23] | 70.19 ± 0.23 | 86.10 ± 0.14 | 65.22 ± 0.23 | 82.20 ± 0.15 | 79.06 ± 0.20 | 90.92 ± 0.11 | 80.46 ± 0.21 | 93.83 ± 0.09 |
| +S2 [43] | 71.18 ± 0.23 | 84.16 ± 0.15 | 66.51 ± 0.24 | 80.32 ± 0.17 | 77.38 ± 0.21 | 88.78 ± 0.12 | 81.97 ± 0.20 | 91.09 ± 0.12 |
| +GAM [20] | 69.16 ± 0.23 | 84.56 ± 0.14 | 64.20 ± 0.24 | 80.09 ± 0.16 | 77.96 ± 0.20 | 90.62 ± 0.11 | 81.22 ± 0.21 | 93.63 ± 0.09 |
| +Parnet [6] | 70.42 ± 0.22 | 86.39 ± 0.13 | 65.93 ± 0.22 | 83.03 ± 0.15 | 76.19 ± 0.21 | 90.59 ± 0.11 | 82.06 ± 0.20 | 93.71 ± 0.09 |
| +ACmixAttention [25] | 69.17 ± 0.22 | 85.58 ± 0.14 | 67.35 ± 0.22 | 83.17 ± 0.14 | 76.25 ± 0.21 | 90.39 ± 0.11 | 82.43 ± 0.20 | **94.16 ± 0.09** |
| +DAN [37] | 62.59 ± 0.23 | 84.06 ± 0.15 | 66.81 ± 0.22 | 82.94 ± 0.14 | 79.74 ± 0.20 | 90.65 ± 0.12 | 83.35 ± 0.21 | 93.06 ± 0.10 |
| **+CSCAM (Ours)** | **72.33 ± 0.21** | **87.18 ± 0.13** | **67.58 ± 0.23** | **83.25 ± 0.15** | **81.69 ± 0.20** | **91.01 ± 0.11** | **85.73 ± 0.18** | 93.26 ± 0.10 |

Additionally, we conduct experiments using different backbone architectures. Specifically, we use the Conv-4 backbone [13, 32] and the ResNet-12 backbone [8, 13]. For the Conv-4 backbone, we set the parameter training way to 20, while for the ResNet-12 backbone, We set the parameter training way to 10.

## 4.3 Comparison with State-of-the-Art Methods

In our experiments, we evaluate the performance of the proposed method and compared it against state-of-the-art metric-based few-shot methods on four fine-grained benchmark datasets in Table 1.

As shown in Table 1, when integrated into Proto [28], Relation [30], FRN [38] and TDM [14], our method can mostly improve their performances, on both ResNet-12 and Conv-4 backbones. In particular, when the Conv-4 backbone is used, CSCAM integrated with FRN or TDM achieves the highest accuracy on all four datasets; when the ResNet-12 backbone is used, CSCAM integrated with FRN or TDM achieves the highest accuracy on three datasets. This verifies general effectiveness of our CSCAM as a plug-and-play module for various few-shot fine-grained image classification models.

## 4.4 Comparison with Other Attention Modules

To evaluate the effectiveness of our method in comparison to other attention mechanisms, we integrate various attention modules into Proto [28] and list their performances in Table 2. The compared attention modules include those of SE [10], CBAM [39], DA [5], CAM [9], Triplet [23], S2 [43], GAM [20], Parnet [6], ACmixAttention [25] and DAN [37], as presented in their official papers. We only take their attention modules to compare with the proposed CSCAM and do not use the whole network of these methods. We note that some of the previously mentioned attention methods [4, 42, 47] have no officially published code, hence we only compare their entire network with our proposed method in Table 1.

As shown in Table 2, CSCAM has the highest classification performance in seven out of eight few-shot settings. This shows that our CSCAM, which exploit cross-attentions from both channel and spatial perspectives and can generate more discriminative features, is superior to these current attention mechanisms for few-shot fine-grained image classification.

## 4.5 Ablation Studies

To further explore the effectiveness of two sub-attention modules and the support averaging method of our CSCAM, we conduct a series of ablation studies: (1) what if without the spatial cross-attention module (SCAM); (2) what if without the channel cross-attention module (CCAM); and (3) what if without the support averaging method. We perform the ablation experiments for both FRN and Proto. The experimental results are presented in Table 3.

From the results, we can observe the followings. Firstly, using SCAM or CCAM alone with the support averaging can already improve the baseline performance. Secondly, if integrated into FRN, CCAM performs mostly better than SCAM. However, if integrated into Proto, CCAM can be often worse than SCAM. This suggests that each of spatial cross-attention and channel cross-attention has their own strength. Thirdly, the proposed CSCAM, which combines both CCAM and SCAM, always performs the best. This proves that the attention mechanism from a single perspective is insufficient, and the best performance improvement can be achieved by leveraging cross-attentions collaboratively from both spatial and channel perspectives. Finally, a comparison of the experimental results in the last two rows of each table shows the positive effect of the support averaging method.

## 4.6 Visualization

*4.6.1 Feature visualization by Grad-CAM.* To verify the effectiveness of CSCAM in learning discriminative features, we utilize Grad-CAM [26] to visualize the features. The visualization results are presented in Figure 5, where the heatmaps highlight the regions that contribute significantly to the classification decisions.

We can observe the following patterns. First, it shows that in some cases, Proto [28] and FRN [38] focus on background regions (e.g., the sky in the Aircraft dataset, the grass in the Flowers dataset, the wall in the Cars dataset). Secondly, in comparison with Proto and FRN, when the proposed sub-module CCAM or SCAM is integrated into FRN, the model can focus more on regions critical for classification. Thirdly, when CSCAM, which includes both CCAM and SCAM, is integrated into FRN, the model accurately focuses on the discriminative regions of fine-grained images (such as the claws,

**Table 3: Ablation studies on the CUB, Flower, and Cars datasets, employing the Proto (Conv-4) and FRN (Conv-4) backbones in a 5-way few-shot setup. These studies involved the removal of both CCAM and SCAM, as well as the separate utilization of CCAM or SCAM.**

| Proto [28] | | | CUB | | Flowers | | Cars | |
|---|---|---|---|---|---|---|---|---|
| CCAM | SCAM | SA | 1-shot | 5-shot | 1-shot | 5-shot | 1-shot | 5-shot |
| × | × | × | 61.82 ± 0.23 | 83.37 ± 0.15 | 64.23 ± 0.23 | 84.94 ± 0.16 | 48.42 ± 0.22 | 71.38 ± 0.18 |
| ✓ | × | ✓ | 71.17 ± 0.22 | 85.72 ± 0.14 | 69.77 ± 0.23 | 84.07 ± 0.17 | 65.73 ± 0.22 | 82.82 ± 0.16 |
| × | ✓ | ✓ | 71.67 ± 0.22 | 86.08 ± 0.14 | 70.24 ± 0.22 | 85.12 ± 0.16 | 64.30 ± 0.23 | 81.61 ± 0.16 |
| ✓ | ✓ | × | 71.94 ± 0.23 | 86.89 ± 0.14 | 71.13 ± 0.21 | 85.28 ± 0.15 | 66.13 ± 0.23 | 81.52 ± 0.16 |
| ✓ | ✓ | ✓ | **72.33 ± 0.21** | **87.18 ± 0.13** | **71.20 ± 0.23** | **85.43 ± 0.16** | **66.58 ± 0.23** | **83.25 ± 0.15** |

| FRN [38] | | | CUB | | Flowers | | Cars | |
|---|---|---|---|---|---|---|---|---|
| CCAM | SCAM | SA | 1-shot | 5-shot | 1-shot | 5-shot | 1-shot | 5-shot |
| × | × | × | 73.46 ± 0.21 | 88.13 ± 0.13 | 73.60 ± 0.22 | 88.69 ± 0.14 | 64.03 ± 0.22 | 84.02 ± 0.13 |
| ✓ | × | ✓ | 76.34 ± 0.21 | 89.28 ± 0.13 | 74.16 ± 0.22 | 88.17 ± 0.14 | 71.40 ± 0.22 | 86.17 ± 0.13 |
| × | ✓ | ✓ | 76.59 ± 0.21 | 88.88 ± 0.12 | 72.89 ± 0.22 | 87.53 ± 0.15 | 70.76 ± 0.22 | 85.74 ± 0.13 |
| ✓ | ✓ | × | 77.26 ± 0.21 | 89.38 ± 0.12 | 73.68 ± 0.22 | 88.06 ± 0.16 | 70.34 ± 0.23 | 85.82 ± 0.16 |
| ✓ | ✓ | ✓ | **77.68 ± 0.20** | **89.88 ± 0.12** | **74.29 ± 0.22** | **88.70 ± 0.14** | **71.44 ± 0.21** | **86.44 ± 0.13** |

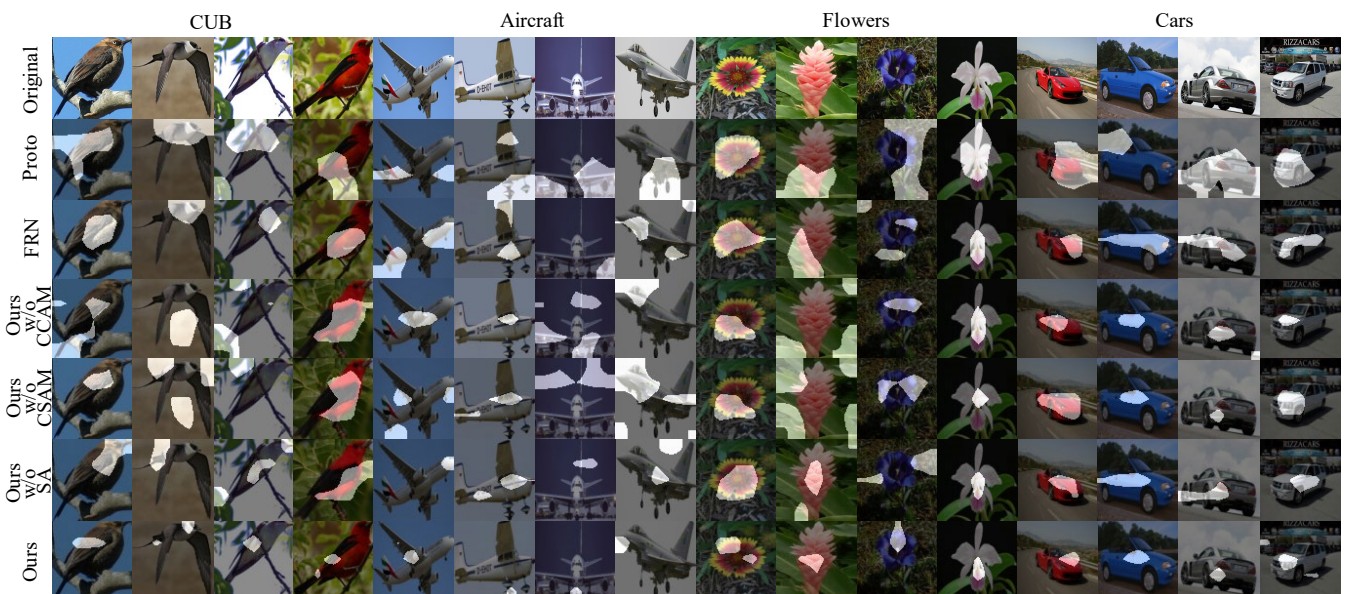

**Figure 5: Visualization of features extracted by Proto, FRN, and our proposed method and its ablated variants, using Grad-CAM [26] for the visualization on the CUB, Aircraft, Flowers, and Cars datasets.**

beak, and wings of birds; the head and engine of aircraft; the petal, core, and edge of flowers; the lights, mirrors, roof, and handlebars of cars). This indicates that the attention mechanism of CSCAM successfully captures discriminative features and effectively attends to the critical parts of the input images for classification.

*4.6.2 Feature visualization by T-SNE.* Furthermore, we will use T-SNE [33] to illustrate the effect of support averaging on increasing the inter-class distance and decreasing the intra-class distance.

Specifically, we take ResNet-12 as the backbone and compare the visualization results of their 5-way 5-shot on two fine-grained benchmarks, CUB and Cars, as shown in Figure 6, in which the same color represents the same class of samples.

From Figure 6, by comparing the proposed support averaging method (*Ours*) with the case without using it (*w/o SA*), it can be found that the support averaging method can enlarge the distance between feature classes while reducing the distance within a class, which verifies the rationality and positive effect of the support

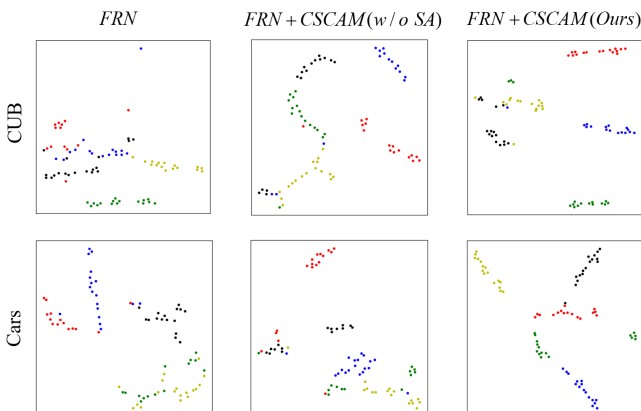

**Figure 6: Features dimension-reduction visualization by T-SNE for *FRN, FRN+CSCAM(w/o SA)*, and *FRN+CSCAM (Ours)* on CUB and Cars with ResNet-12 for 5-way 5-shot.**

averaging method. In addition, by comparing the feature dimension-reduction visualization of *FRN + CSCAM* and *FRN*, it can be found that the between-class distance of *FRN + CSCAM* is clearer and more separable, and the within-class distance is relatively more compact, which confirms that the proposed method can better meet the requirements of fine-grained image classification and is more suitable for the few-shot fine-grained classification tasks.

## 4.7 Remarks on Strength and Limitation

*4.7.1 Strength.* Through the above experimental comparisons and visualizations, it can be verified that our method can improve state-of-the-art performance in few-shot fine-grained image classification. This can be ascribed to several reasons. Firstly, CSCAM makes full use of the channel and spatial cross-attention between query features and support features, hence the network can obtain richer information than the existing attention modules, so as to improve the classification of fine-grained images. Secondly, while focusing on the intra-class similarity, the proposed CSCAM also considers the regions with high inter-class similarity by averaging the attention scores over the entire support set through the support averaging method, which can further expand the inter-class distance and reduce the intra-class distance, so as to generate more fine-grained and discriminative features. It hence can better meet the needs of few-shot fine-grained image classification tasks.

*4.7.2 Limitation.* Although in the task of few-shot fine-grained image classification, the proposed CSCAM can achieve a good performance improvement when inserted into the existing metric-based few-shot learning models, it still has the following limitation: In the attention computation, the space overhead is large. Hence it is our future work to reduce various computational overheads in the attention computation.

## 5 CONCLUSION

In this paper, we proposed a channel-spatial cross-attention module (CSCAM) for few-shot fine-grained image classification. The proposed attention module contains a channel cross-attention module and a spatial cross-attention module. Different from other self-attention methods based on single spatial or channel cross-attention in few-shot learning, CSCAM captures the cross-attention between query features and support features from both channel and space simultaneously. In addition, the support averaging method proposed to fit for the characteristics of fine-grained images can enlarge the between-class distance and reduce the within-class distance, obtaining more subtle and discriminative features for fine-grained image classification. Extensive experiments show that CSCAM can clearly improve state-of-the-art performance on few-shot fine-grained image classification.

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
