# OpenReview forum: "Channel-Spatial Support-Query Cross-Attention for Fine-Grained Few-Shot Image Classification"
_acmmm.org/ACMMM/2024/Conference — MM2024 Poster_

### Official Review · Reviewer_YLZk · 2024-05-14

**Rating:** 4
**Confidence:** 2

**Summary:**

For the task of fine-grained few-show image classification, a channel-spatial cross attention module (i.e. CSCAM) has been proposed in this paper. The cross-attention between query features and support features from both channel and space can be captured by the CSCAM, simultaneously. Moreover, a support averaging method is also proposed in CSCAM to reduce the intra-class distance and increase the inter-class distance according to the characteristics of fine-grained images. The CSCAM can be considered as a plug-and-play module for integrating into various existing metric-based few-shot methods.

**Strengths:**

The proposed CSCAM is taken as a plug-and-play module for integrating into various existing metric-based few-shot methods. Based on effective evaluation, when the CSCAM is integrated into state-of-the-art metric-based few-shot methods, the performance can be improved. In addition, the CSCAM is superior to other attention modules by comparisons.

**Limitations:**

(1) In Table 1, the results of BSNet (D&C) [17] on the dataset of Aircraft are wrong. The corresponding results are 62.86% (1-shot) and 83.12% (5-shot), respectively. Please check up. Additionally, the results of MattML [47] on the dataset of Aircraft are not listed.
(2) In Table 2, various attention modules are compared on only two datasets. The other two datasets (i.e. Aircraft and Flowers) are not verified, why?
(3) In subsection 4.3, the proposed CSCAM module were integrated into Proto [28], Relation[30], FRN [38] and TDM [14]. Why the CSCAM is not integrated into much more metric-based few-shot methods (e.g. MattML [47])?

**Suitability:**

3

---

### Official Review · Reviewer_PBSe · 2024-05-23

**Rating:** 3
**Confidence:** 3

**Summary:**

This paper proposes a Channel-Spatial Cross-Attention Module which integrates a channel cross-attention module and a spatial cross-attention module for the task of few-shot fine-grained image classification. The experimental results of the proposed method on four fine-grained datasets (CUB-200-2011, FGVC-Aircraft, Flowers-102, and Standford-Cars) are better than those of the comparison methods.

**Strengths:**

The proposed method considers the cross-attentions from both channel and space perspectives, leveraging the spatial and channel cross-information between support and query.

**Limitations:**

1.	The two innovation points proposed in this paper are relatively weak. The description of research motivation in the paper is not clear, so it is suggested to explain it clearly in the sections of abstract and introduction. What the difficulties can the channel-spatial cross-attention module solve in the task of fine-grained few-shot image classification?
2.	Since the proposed CSCAM is plug-and-play, why does this method use the previous methods (such as Proto, FRN and TDM) as the backbone instead of the latest methods (such as C2-Net and BIFRN) in Table 1?

**Suitability:**

3

---

### Official Review · Reviewer_2TSy · 2024-05-24

**Rating:** 4
**Confidence:** 4

**Summary:**

This paper proposes a channel-spatial cross-attention module to tackle the few-shot fine-grained image classification problem, which is vaulable and meaningful for practical application. This paper utilizes the channel-wise attention and spatial-wise attention, which is different from previous attention-based works in few-shot fine-grained community and shows promising classification performance gains.

**Strengths:**

- This paper is easy to follow and well organized.
- Experimens in this paper cover many benchmarks including CUB, Aircraft, Flowers, Cars, and many baseline models.
- The few-shot performance gain is promising in both 1-shot and 5-shot setting.
- The authors give detailed discussion including visual results and overall strength in experiment section, showing the deep insight for authors.

**Limitations:**

- The technical contribution of this paper should be enhanced. The core technical contribution is the proposed Channel-Spatial Cross-Attention Module (CSCAM). It leverages cross-attention structure to encode the relation along the channel-level and spatial-level, what is the differnce of such spatial attention between this work and [Ref A]. Please enhance the technical contribution and explain the differences during the rebuttal phase.

[Ref A] Object-aware Long-short-range Spatial Alignment for Few-Shot Fine-Grained Image Classification. ACM MM-2021.

- For CCAM that encodes the channel-level attention, will the approach lose the correlation between different elements within the channel dimension, since the features distribution along channel dimension is important for fine-grained classification. Please provide some explainations and discussions.

**Suitability:**

3

---

### Official Review · Reviewer_W4u6 · 2024-05-24

**Rating:** 5
**Confidence:** 4

**Summary:**

This paper introduces a channel-spatial cross-attention module, comprising channel cross-attention and spatial cross-attention submodules, which apply attention weighting to query features from two perspectives. It also proposes an average support method to decrease intra-class distances and increase inter-class distances. Extensive experiments on multiple fine-grained datasets validate its effectiveness and competitiveness.

**Strengths:**

The method presented in this paper is competitive in the domain of fine-grained image classification for small sample sizes, compared to many advanced methods. The proposed average support method's effectiveness is also substantiated through ablation studies. This method, as a plug-in, adapts well to few-shot fine-grained image classification tasks.

**Limitations:**

1. The ablation experiments for the two sub-attention modules and the average support method are not comprehensive; further experiments are recommended.
2. Figure 6 features a limited dataset for visualization, lacking sufficient persuasiveness; adding more datasets' visual results is suggested.
3. The explanations for Eq. (3) and Eq. (6) regarding the average support method are unclear; a clearer explanation is necessary.
4. Figure 5 in the paper is overly congested; it is advised to split this into Figures 5a and 5b for a clearer display.

**Suitability:**

2

---

### Meta-Review · Area_Chair_nEbs · 2024-06-29

**Recommendation:** Accept (Poster)
**Confidence:** 4

**Metareview:**

The paper proposes a channel-spatial cross attention module (CSCAM) for fine-grained few-shot image classification. CSCAM captures cross-attention between query and support features in both channel and spatial domains. It includes a support averaging method to minimize intra-class and maximize inter-class distances, enhancing fine-grained image classification. Designed as a plug-and-play module, CSCAM can easily integrate into existing metric-based few-shot methods.